# The Effect of Participative Leadership Style on Employees' Performance: The Contingent Role of Institutional Theory

**Osama Khassawneh** [1,*] and **Hamzah Elrehail** [2]

1   Lazaridis School of Business and Economics, Wilfrid Laurier University, Waterloo, ON N2L 3C5, Canada
2   Department of Leadership and Organizational, Abu Dhabi School of Management,
    Abu Dhabi 999041, United Arab Emirates
*   Correspondence: khasawneho@hotmail.com

**Abstract:** This study aimed to examine the moderating role of institutional theory in the association between participative leadership style and various outcomes, such as employee loyalty and job performance in organizations. A cross-sectional research design was employed, where data were gathered from 347 participants from all managerial levels in the United Arab Emirates (UAE). The findings demonstrated how the level of complexity of the institutional theory reduces the positive relationship between participative leadership style and employee loyalty, negatively affecting job performance. The current study contributes to the existing leadership literature by showing that participatory leaders do not behave similarly across various degrees of institutional theory complexity. The findings suggest that the higher the complexity of institutionalism, the wider the gap between leaders and subordinates, so implementing the participative style may become problematic in some circumstances.

**Keywords:** employee loyalty; subordinate; institutional theory; job performance; leader; participative leadership

## 1. Introduction

The participative leadership style demonstrates several conceptualizations, including delegation, joint decision-making, and defined participation. Similarly, Somech (2005) defines participative leadership as making a decision jointly or demonstrating a shared influence in determining superior and subordinate through the hierarchy. As such, the focus of participatory management has become the sharing of power and decision-making allocation. Participative decision-making has been studied as a formal strategy for the direct participation of groups, wherein, in insignificant matters, group participation is considered relevant and influences the group's decisions (Dolatabadi and Safa 2010; Mohammad et al. 2021). Decision-making participation leads to augmented social capacity, with the quality of decisions influencing an increase in employee motivation, work-life quality, the work environment, and professional training in a successful organization (Chan 2019; Ghaffari et al. 2017; Lumbasi et al. 2016). Odoardi et al. (2019) state that the organization and individual outcomes are affected by participative decision-making and this influence can be attributed to augmented employee motivation levels. The quality of decisions is improved through employee participation in the decision-making process, as this helps the supervisor develop an insight into the core issues in a problem situation. Several scholars (Lythreatis et al. 2019; Raineri 2016) argue that this involvement enhances employees' propensity to follow managerial decisions with loyalty. Participative managers value employees' opinions and perspectives and seek their input and suggestions (Rana et al. 2019; Khassawneh and Abaker 2022). Furthermore, participative leaders motivate their employees to develop learning through information acquisition, sharing, and connecting as well as seeking new opportunities (Benoliel and Barth 2017; Mohammad and Khassawneh 2022).

Organizational communication scholars are significantly intrigued by the institutional perspective. According to Lammers and Garcia (2017), institutional theory elucidates the requisite regulations and rules that are necessarily abided by organizations seeking support and legitimacy. This perspective has emerged as a necessary imperative, considering every nation-state, industry, and the various rules and requirements that today regulate the sector. Irrespectively, the paper puts forth a discussion on the critical intersection between institutional theory and organizational communication. According to Cardinale (2018), individuals' communicative behavior in organizations or groups primarily constitutes the focus of corporate communication, that is, their use of language and social interaction, which align coordinated action to achieve a common goal. As such, one can safely deem the larger institutional landscape to fall outside the confines of organizational communication.

Several studies have focused on job commitment, performance, and satisfaction parameters directly resulting from leadership influences (Belias et al. 2022; Budak and Erdal 2022). This paper conducts a descriptive study of the moderating institutional theory in UAE organizations in the context of participative leadership style and several outcomes, including employee loyalty and job performance (Khassawneh 2018; Mohammad 2019).

This study employs an institutional theory to obtain insight into the participative leadership style and how various practice adoptions influence it in the context of organizational performance. According to Bitektine et al. (2018), research focusing on institutional theory is scarce, with nonexistent empirical work despite its significant potential value in behavioral research. Correspondingly, this study attempts to fill this gap by testing the moderating role of institutional theory on the relationship between participative leadership and associated outcomes, such as employees' loyalty and job performance. The paper's ultimate aim and contribution are that a more empirical study of the interplay among institutional theory, participative leadership, employees' loyalty, and job performance is required to verify this premise, which has never been examined in an Arabian context.

## 2. Theoretical Background

According to Bell et al. (2018), when a leader involves and consults with their subordinates to resolve an issue and decide the corrective action, it is referred to as participative leadership and is also referred to as shared influence or joint decision-making (Mwaisaka et al. 2019; Vance 2016), wherein the decision-making process demonstrates the incorporation of the perspectives by the supervisor. Thus, Hayat Bhatti et al. (2019) claim that a supervisor gives subordinates a certain degree of workplace responsibility in this leadership style. Extensive empirical research in diverse cultural and industrial contexts is available, focusing on participative leadership's positive impact on work outcomes (Tang 2019; Fatima et al. 2017; Huang et al. 2006; Somech 2003); more specifically, increased and improved organizational commitment (Salahuddin 2010), voice behavior (Fatima et al. 2017), organizational citizenship behavior (OCB) (Huang et al. 2006), and job performance (Huang et al. 2006)

According to Sax and Torp (2015), the process wherein subordinates are consulted with a focus on their perspective before the leaders' decision-making is termed "participative leadership". Moreover, the concept relates to delegation, consultation, consensus, and involvement (Khassawneh and Abaker 2022; Sarti 2014).

The outcomes related to participative leadership demonstrate that employees show higher organizational commitment, job satisfaction, and performance when they perceive their managers as adopting consultative or participative leadership (Iqbal et al. 2015). Because of the consultative nature of participatory leadership, employees have a greater chance of being exposed to organizational and managerial values. In addition, these employees are inclined toward higher loyalty, commitment, and involvement than employees with a directive leader (Locke and Anderson 2010). Similarly, employees tend to be more committed to decisions when participating in decision-making. Consider, for example, the frontline employees in banking services. Since these workers are in direct contact with customers, they are more cognizant of customer needs than managers. This exam-

ple clearly illustrates the significance of employees' participation in the decision-making process. Managers who aspire to motivate their employees to share their commitment to service quality can benefit from the outcomes of participative leadership, like increased commitment, involvement, and loyalty among employees (Jain and Chaudhary 2014).

Organizational and institutional theories provide a rich and complex view of organizations. Herein, normative pressures influence organizations, which may result from internal and external sources (Heugens and Lander 2009). Moreover, according to Heikkila and Isett (2004), mechanisms and processes such as operating procedures, professional certifications, and state requirements sometimes work as guiding pressures wherein the organization's focus is drawn away from task performance. Because of these legal aspects of adoption, institutional environments develop isomorphism, increasing the likelihood of survival. Correspondingly, Nielsen and Massa (2013) state that the rapid spread of institutional theories of organization evidences the significance of imaginative ideas resulting from theoretical and empirical work. On the same line, an increased number of organizational researchers are expected to be interested in institutional theories, with the better specification of indicators and models concomitant with increased traction on this concept. The institutional theory is complex in a single statement, as it leverages and optimizes the typically neglected assumptions at the core of social action. As such, this paper sheds light on making the institutional theory more accessible (Santos and Eisenhardt 2005). The review is initiated by briefly describing the two current theoretical approaches to institutionalization in organizations, then identifying the central concepts' indicators and transitioning to a review of empirical research. The study culminates with a discussion on the (i) intersection points with other organization theories and (ii) the "new institutionalism" in economics and political science (Khassawneh and Mohammad 2022a; Suárez and Bromley 2016; Huang et al. 2011).

The institutional theory focuses on the more complex and durable facets of social structure and is used in sociology and organizational studies. It views the procedures by which structures, such as plans, regulations, customs, and routines, come to be formed as the supreme standards for social conduct. The creation, diffusion, adoption, adaptation, decline, and disuse of these characteristics over time and space are all explained by various aspects of institutional theory. A developing viewpoint in sociology and organizational studies that (Powell and DiMaggio 1991) refer to as "new institutionalism" rejects the rational-actor models of classical economics. It instead looks for cognitive and cultural explanations of social and organizational events by considering the characteristics of supra-individual units of analysis that cannot be reduced to aggregations or direct outcomes of people's characteristics or motivations.

Let us start by exploring the intersection between participative leadership and employee loyalty, considering the institutional context as a moderator. Several scholars have focused on the association between participative leadership and employee loyalty, inferring a positive intersection, including (Suharti and Suliyanto 2012; Rok 2009; Sorenson 2000).

Participative leadership style claims that participatory leaders tend to focus on the growth and well-being of subordinates, which can be attributed to their sensitivity to subordinate needs. As a result of their interpersonal relationship with their subordinates, leaders influence an increase in employee loyalty.

However, legal and social policies, among other barriers, may disrupt participative leaders' adaptation to subordinates' expectations and needs. The influencing process demonstrates diverse dynamics based on the extent of "high" or "low" policies and regulations in an organizational ecosystem (Khassawneh and Mohammad 2022b; Lok and Crawford 2004). Thus, with restrictions based on policies or corporate cultures, leaders' interaction with their subordinates becomes curtailed. Along the same lines, the degree of interaction between the leaders and the subordinates can be impacted by the institutional context, as leaders can only spend limited time with their subordinates. There are several outcomes associated with stricter institutional factors. For instance, complex institutional factors might limit the ability of participative leaders to show trust in their subordinates;

limited opportunities to respond positively. Moreover, it can limit the scope of leaders' responsibilities to support their subordinates when they experience setbacks (Bitektine and Haack 2015).

The extent of their influence on their subordinates creates a more significant social distance, which can be attributed to leaders' homogeneous treatment of subordinates and less individualized attention (Lok and Crawford 2004). Furthermore, it may limit the ability to help and assist subordinates with setbacks and problems.

These determinants lead to significant differentiation in subordinate membership in either the in-group or the out-group. More specifically, the relationship between participative leaders and subordinates is significantly weak. It only pertains to minimum trust, interaction, and support-based change, which can only be classified as economic change (Mulki et al. 2015). Conversely, a sense of community is built by participatory leaders within more flexible institutional factors by establishing and maintaining a positive working relationship based on trust and caring. It could be claimed that diversity exists even in mature institutional contexts, even though institutions are more likely to support and sustain complex HR systems (Haak-Saheem et al. 2017a).

On the other hand, it may be argued that when institutions are less efficient, firms have more incentives to come up with their solutions (Diab 2022). As a result, it could be argued that combinations of HR practices may compensate for systemic shortfalls; for example, limitations in national training systems may make internal HR development more critical; however, skill shortages may intensify poaching unless firms doing the training and development devise supportive reward systems to retain the people they have invested in. In turn, such behaviors can be copied, providing tried-and-true recipes for succeeding in challenging situations. In other words, groups of HR practices may produce better results in the UAE (Haak-Saheem et al. 2017b; Powell and Colyvas 2008). Hence, we suggest the following hypothesis (see Figure 1):

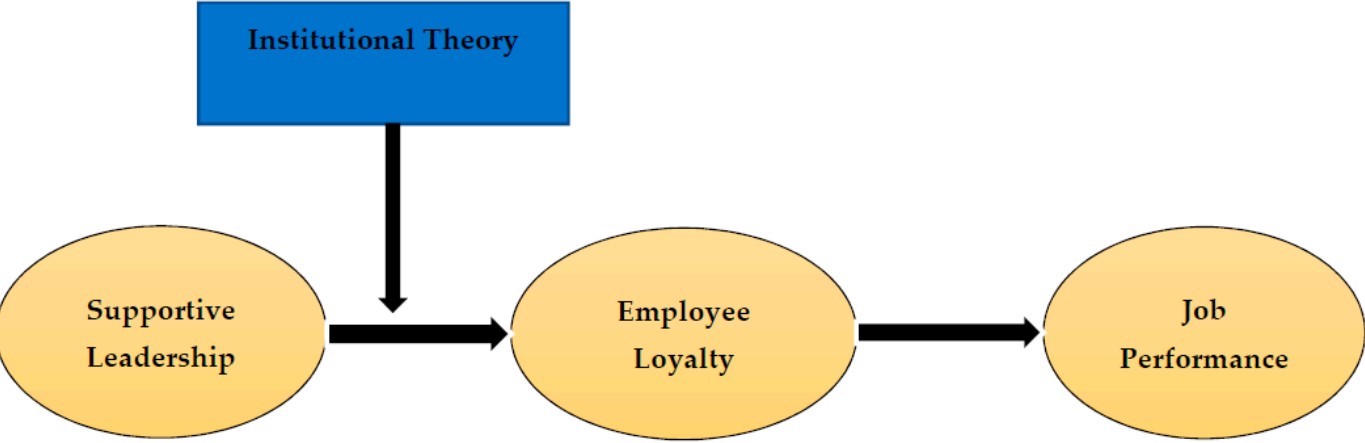

**Figure 1.** Proposed Theoretical Model.

**Hypothesis 1.** *More complex institutional context attenuates the positive relationship between participative leadership and employee loyalty.*

According to Cook et al. (2013) social exchange theory, subordinate job loyalty should be accompanied by an increased subordinate performance at an organization. In the context of subordinate performance, the leader and the organization are positively impacted by augmented employee loyalty. Several empirical studies have inferred a significant positive association between employee loyalty and job performance. For example, Joshi et al. (2015) conducted a meta-analysis on 190 samples with a combined N of 64.516 and estimated the mean correlation between employee loyalty and job performance to be 0.40. According to Cooper et al. (2019), the correlation between the two variables in a more recent meta-analysis of 69 samples was inferred at 0.40.

Moreover, the broader social psychology literature also highlights a positive association between employee loyalty and work performance. According to Fleischman et al. (2017), employees deemed to have a negative attitude are inclined to engage in behaviors that oppose it, and vice versa.

Thus, the study proposes a positive association between employee loyalty and job performance.

**Hypothesis 2.** *Employee loyalty is associated with higher levels of subordinate performance.*

## 3. Methodology

### 3.1. Sample and Participants

The study participants were full-time employees of large organizations in the UAE. We distributed the survey to 32 companies in the UAE. These companies run businesses in the service industry, including eight banks (88 employees), six hospitals (76 employees), ten hotels (96 employees), and eight insurance companies (87 employees). We targeted 500 employees but were able to collect 347 responses. The respondents were full-time employed. The data were obtained from leaders and subordinates at various organizational levels. We chose the service sector because the UAE is heavily reliant on it. The leader sample entailed 67 members, of whom 34.5% were males. The leaders' mean age was 41.6 years, with an average education of 15.7 years. They were hired as general managers, directors, area managers, department managers, assistant managers, supervisors, and team leaders. The subordinate sample entailed 280 participants, of whom 34.8% were males. Their mean age was 34.4 years, and their average education level was 15.7 years. They perform different jobs: administrators, customer service representatives, accountants, HR officers, and bookkeepers.

During the study, we were given access to the organization's structure, study participants, and their contact details, i.e., respondents' email addresses. For ethical reasons, the identity of the respondents was maintained anonymous, and completed questionnaires (82.6%) were returned directly to the researchers. An electronic medium was used for data collection to share the questionnaires during work hours, and informed consent was included in the questionnaire.

### 3.2. Instruments

With this study, we conducted a pilot study to ensure that the study methodology aligns with the study focus. In the pilot study, we evaluated the instruments, the distribution of questionnaires, and the data collection procedure. After this, minor alterations were made to the tools before they were shared with the study participants.

To assess the work performance, each supervisor considered the subordinates using a five-item performance rating scale developed by (Liden and Graen 1980) (sample items: "Overall Present Performance" and "Future Expected Performance", anchors: 1 = unsatisfactory 7 = outstanding). The survey included the overall performance indicators (e.g., the speed of getting the task done, the customer satisfaction level, the number of functions, and complaints). The collected responses to these five items were summed to offer a performance measure for each subordinate. Moreover, to reduce the common method bias that could probably result from self-reported measures, we used two sources (i.e., leader and subordinate self-report) on the ratings.

### 3.3. Institutional Factors

The leaders reported the volume and degree of restriction of institutional factors for their companies. Each subordinate finished the following instruments: This 30-item version of the participatory leadership survey, based on the format proposed by (Ramli and Desa 2014), was used to assess the eight parameters of standing back: forgiveness, courage, empowerment, accountability, authenticity, humility, and stewardship. The subordinates were asked to share their responses on a five-point Likert-type scale, ranging from 1

(strongly disagree) to 5 (strongly agree). Scholars to date have developed at least seven multidimensional and two one-dimensional measures, such as those developed by (Park et al. 2016; Ogbeide and Harrington 2011; Elele and Fields 2010), for participative leadership. However, as regards their robustness, several scholars (e.g., Khassawneh 2018; Stirna et al. 2007; Parnell et al. 2002) have argued that instruments constructed to measure the multidimensional structure of participative leadership are collapsed into one and do not hold across several samples. An employee loyalty questionnaire, based on the format propounded by (Yee et al. 2010), was used to measure loyalty via a three-item scale: "All in all, I am loyal to my job", "In general, I like to stay here", anchors: 1 = strongly disagree 2 = disagree somewhat 3 = slightly disagree 4 = neither agree nor disagree 5 = slightly agree 6 = agree somewhat, and 7 = strongly agree. Overall, employee loyalty was measured using the index. Several studies have reported the adequacy of the employee loyalty scale in reliability and validity analysis (Rice et al. 2017; Khassawneh and Mohammad 2022a).

### 3.4. Control Variables

In the current paper, the control condition used was gender. According to Bernerth et al. (2018), female leaders are expected to be more understanding, helpful, sophisticated, and sensitive to others' feelings. The other control variable was age, attributed to Newey and Stouli (2018) inference that younger supervisors show higher engagement in relationship-oriented activities than older supervisors. Another control variable was education, as underpinned by Nielsen and Raswant (2018), who found that a more personal, individualized, and cooperative leadership style is displayed in individuals with higher educational qualifications. We also considered the years of experience as a control variable. However, there was no relationship between the years of experience and the leadership style. Moreover, the analysis controlled for institutional factors and participative leadership when testing the hypothesized relationship between employee and job performance, exploring alternative explanations for the relationships outlined in our hypotheses.

### 3.5. Confirmatory Factor Analysis

MPlus was used to test the degree of match or alignment between the predicted interrelationships and the variables with the interrelationships between the observed interrelationships to estimate the confirmatory factor analysis (CFA). The CFA results inferred the following results: CFA provided an excellent fit to the data ($\chi^2$ (682) = 1037.55, $p < 0.05$; RMSEA = 0.03; CFI = 0.95; NNFI/TLI = 0.96). Corresponding to the results of (Keith and Reynolds 2018), CFA demonstrated an excellent model fit compared to the results with frequently used rules of thumb.

## 4. Results

Table 1 presents the descriptive statistics. The results show a significant and positive relationship between participative leadership and employee loyalty (R = 0.39, $p < 0.01$) and a positive co-relationship between employee loyalty and job performance (R = 0.17, $p < 0.05$). Cronbach's multi-item scales are listed on the primary diagonal of the correlation matrix. The $\alpha$ coefficients fell into an acceptable range for all the variables of interest (0.74 to 0.95).

**Table 1.** Descriptive statistics and correlations.

| Variables | Means (SD) | 1 | 2 | 3 | 4 | 5 | 6 | 7 |
|---|---|---|---|---|---|---|---|---|
| Years of Education for Leaders | 15.7 (1.80) | | | | | | | |
| Age of Subordinate | 41.6 (7.74) | 0.33 ** | | | | | | |
| Gender of Leader [a] | 0.69.9 (0.57) | −0.18 ** | 0.03 | | | | | |
| Participative Leadership | 3.72 (0.63) | 0.08 | 0.05 | 0.07 | (0.91) | | | |
| Institutional Context | 13.55 (8.08) | 0.40 ** | −0.02 | 0.05 | −0.07 | | | |
| Employee Loyalty | 7.14 (1.09) | 0.06 | −0.02 | −0.03 | 0.39 ** | −0.00 | (0.74) | |
| Job Performance | 6.31 (1.15) | 0.27 ** | 0.17 * | 0.09 | 0.28 ** | 0.15 ** | 0.17 * | (0.95) |

* $p < 0.05$; ** $p < 0.01$; [a] 0 = Male 1 = Female; Note: N = 267. Cronbach's as is displayed on the primary diagonal.

*Hierarchical Linear Modeling Analyses*

The nested nature of the study data (i.e., individuals nested within leaders) necessitated testing our hypotheses using hierarchical linear modeling (HLM). Before advancing with these analyses, we estimated unconditional models (null models) for employee loyalty and job performance. The results did not indicate significant between-supervisor variability in employee loyalty; however, they did reveal significant between-supervisor variability in supervisor ratings of performance ($\tau 00 = 0.26$, $p < 0.01$), thus underscoring the appropriateness of HLM. We present the results of these analyses in Tables 2 and 3. In step 1 of Table 2, we entered the independent variable, participative leadership. In step 2, we entered the moderating variable, institutional context. In step 3, we entered the product terms "participative leadership" and "institutional context". According to the findings, the relationship between participative leadership and employee loyalty was significantly ($=-0.04$, $p < 0.05$) moderated by institutional context and this confirm hypothesis 1. We used the HLM two-way interaction tool to determine the significance of the simple slopes. The findings show a positive relationship between participative leadership and employee loyalty only when institutional context constraints are lower ($=-1.07$, $p < 0.001$). In contrast, the relationship with a higher degree of institutional context restrictions was not statistically significant ($=0.40$, n.s.), implying that institutional context represents a boundary condition under which participative leadership relates to employee loyalty. Finally, the results in Table 3 support hypothesis 2 by indicating a positive relationship between employee loyalty and job performance ($=0.18$, $p < 0.05$). Concerning the control variables, we note that, as shown in Table 3, neither the leader's years of education ($=0.01$, n.s.), the subordinate's age ($=0.00$, n.s.), nor the leader's gender ($=-0.07$, n.s.) were significantly related to employee loyalty and this confirm hypothesis 2.

**Table 2.** Results of hierarchical linear modeling analyses.

| Variables | Employee Loyalty | | |
| --- | --- | --- | --- |
| | Step 1 | Step 2 | Step 3 |
| Intercept | 5.28 *** | 5.28 *** | 5.27 *** |
| Years of Education for Leader | 0.02 | 0.01 | 0.01 |
| Age of Subordinate | −0.00 | −0.00 | −0.00 |
| Years of Experience | 0.01 | 0.02 | 0.02 |
| Gender of Leader [a] | −0.08 | −0.08 | −0.07 |
| Participative Leadership | 0.80 *** | 0.80 *** | 0.77 *** |
| Institutional Context | | 0.00 | 0.01 |
| Participative Leadership X Institutional Context | | | −0.04 * |
| Model deviance $\chi^2$ | 388.94 | 388.91 | 383.96 |
| Decrease in Deviance: $\Delta\chi^2$ [b] | | 0.03 | 5.75 * |

* $p < 0.05$; *** $p < 0.001$; [a] 0 = Male 1 = Female; [b] The full ML estimator was applied to compute this decline in deviance. ($\Delta\chi^2$) This can be measured by stating effect size in multi-level modeling. Note: N = 267. Non-standardized coefficients are displayed.

**Table 3.** Results of hierarchical linear modeling analyses.

| Variables | Job Performance |
| --- | --- |
| Intercept | 5.94 *** |
| Years of Education for Leader | 0.14 * |
| Age of Subordinate | 0.00 |
| Years of Experience | 0.02 |
| Gender of Manager [a] | 0.08 |
| Participative Leadership | 0.18 |
| Institutional Context | 0.02 |
| Employee Loyalty | 0.18 * |
| Model Deviance $\chi^2$ | 357.73 |

* $p < 0.05$; *** $p < 0.001$. [a] 0 = Male 1 = Female; ($\Delta\chi^2$) can be measured to state effect size in multi-level modeling. Note: N = 267. Non-standardized coefficients are displayed.

## 5. Discussion

Improved employee loyalty can lead to higher job performance levels. The current paper explored the moderating role of the institutional context of multifocal effectiveness outcomes in participative leadership. The HLM analysis results show that institutional context significantly moderates the relationship between participative leadership and employee loyalty, and this was evident through the weakening impact of higher institutional context levels on the participative leadership and employee loyalty relationship (Zijl et al. 2021). Instead, fewer institutional contexts demonstrated significant relationship levels between participative leadership and employee loyalty (Chang et al. 2021). It was only in situations when the leader had fewer institutional contexts. A likely consequence of employee loyalty is, in turn, higher levels of job performance from the subordinates (Khassawneh et al. 2022; Wang et al. 2022; Pollermann and Fynn 2021).

### 5.1. Theoretical and Practical Implications

Organizational context is a significant variable influencing behavior at the workplace (Ngugi 2019) and leadership behavior and outcomes, specifically (Palihakkara and Weerakkody 2019). When speaking to leaders in large organizations in the UAE, we have learned that a significant institutional context characterizes this type of organization. Institutional context causes distance between leaders and subordinates and limits the participative leaders' possibility to influence their subordinates, probably due to the inability to support associates. As stated by Gandolfi and Stone (2018), a leader's effectiveness is contingent on matching the degree of support that aides expect of their leader. Furthermore, leaders will enact different behaviors depending on the context in which those behaviors occur, Gandolfi and Stone (2018). Hence, institutional context limits participative leaders' possibility to energy on coaching subordinates for innovative performance or providing them with essential support. Restricted institutional context represents an obstacle when implementing participative leadership in organizations and is detrimental to leader outcomes.

Arguably, the competence and commitment of the subordinate may influence the obstructive impact of the institutional context for participative leadership. According to Kimura and Nishikawa (2018), in an organization, the need for a leader's support/supervision is governed by the availability and access to several organizational systems and processes, as well as the subordinate's competence and commitment. Thus, depending upon these leader "substitutes", leadership may be unnecessary. This argument is supported by Opeke and Oyerinde (2019). In their research on situational leadership theory, they state that for a competent and motivated subordinate, delegating leadership style is favorable. In addition, participative leaders may be accorded opportunities for managing larger institutional contexts when assistants conduct simple, repetitive tasks. As such, the competence and commitment of the subordinate should constitute the focus of future research in line with task characteristics to explore, under such conditions, the suitability of participative leadership.

Moreover, the moderating role of institutional context has been studied in this paper. It has a specific relevance with its emerging significance. An increasing focus on policies, rules, and other social settings in organizational settings positively impacts the augmentation of institutional context (see Mohammad and Khassawneh 2022; Peters 1999). The study infers that the larger institutional context adversely impacts the positive intersection of participative leadership with employee loyalty, resulting in poor employee performance. On these lines, the current paper studies the presumed effectiveness of several contemporary organizational change processes, such as reducing restrictions in the institutional context for leaders.

Considering the intersection between participative leadership and employee loyalty, the trends identified in this explorative paper bring an essential boundary condition to the forefront. It reveals that only in conditions demonstrating less institutional context the leader shows increased loyalty among the subordinates in participative leadership, thus suggesting better efficacy of limited institutional context with more flexibility to the participative leader to provide focused support for their needs.

In other words, a narrower institutional context would allow for: (i) higher involvement of subordinates in decision-making, (ii) better explanation of organizational decisions, and organization expectations. Correspondingly, the HLM analysis illustrated a proportional relationship between participative leadership, employee loyalty, and job performance. However, according to Currie et al. (2009), in scenarios wherein there is no opportunity for a decrease in an institutional context, the subordinates can be supported by the participative leaders. Ensuring the availability of increased opportunities for the leader to support subordinates in participative leadership, with a narrow institutional context, can result in improved employee performance that can be attributed to competency enhancement through coaching and feedback (Khassawneh and Mohammad 2022b). Furthermore, distinct roles can be accorded to the subordinates by their participative leaders, allocating variable service diverse types to their coworkers, like peer recognition. The OCB concept propounded this approach and can improve coworkers' performance (Mohammad et al. 2021; Jung and Yoon 2012; Cortes and Herrmann 2021). This approach can also increase the capacity of participative leadership to support more prominent subordinates.

*5.2. Limitations and Directions for Future Research*

This paper provides a robust understanding of the study focus, as we sourced work performance ratings from diverse containers for the parameters of (i) supervisory rating of subordinate's work performance, (ii) subordinate rating of participative leadership, and (iii) employee loyalty. The data were studied to identify trends by cross-referencing with the third data type of institutional context. Thus, as Spector (2006) supported, the multi-source data application ensured that the common method variance effect was minimized. In addition, the standard method bias can be minimized with a guarantee of anonymity, as per Du et al. (2005). The same was accorded to all the study participants, including the leaders and subordinates.

The current paper presents a limitation of reliance on a cross-sectional measurement design. The survey methodology, concentrated on a single time for surveying a significant number of leaders and subordinates, tested the validity of our research model. This approach limited causal inference of causal relationships, excluding alternative causal ordering. This phenomenon has been observed in a previous study conducted by Guillon and Cezanne (2014), which showed that subordinate employee loyalty is associated with work performance. Irrespective, the inference is aligned with the research focus, thereby suggesting that a subordinate's commitment underpins performance, contrasting the hypothesis that organizations are motivated by a subordinate's performance to invest in gaining his loyalty (Ineson and Berechet 2011). Correspondingly, it is recommended that longitudinal data should be applied in future research to facilitate an improved evaluation of the impact that participative leadership has on work performances and related aspects. It can be attributed to the fact that the manifestation of participative leadership is known to be time-consuming.

The current study presents another limitation: most participants were women, even though the survey inferred a relatively high response rate. Thus, the exclusive reliance on a UAE sample with 60% women may limit the generalizability of the results. At the same time, it is also essential to note that the control variable gender was not related to any of the outcomes. In addition, the results were achieved in a particular cultural environment (UAE organizations) and might need help to be easily transferred.

It is recommended that future research assess and subsequently ensure that the research model is generalizable and applicable to cross-cultural contexts. Therefore, it becomes essential that future studies consider multiple and disparate factors, including culture, gender, and organizational type spanning across geographies, economies, and industries. Moreover, it is suggested that different gender distribution ratios should be considered for better generalizability of our hypothesized relationships. As such, it would be interesting to observe the outcomes of studies focusing on hypothesized relationships at the group level through experimental and longitudinal data within the context of cross-

cultural scopes. Future studies would be worth looking at leaders' cultural backgrounds and investigating how this could affect the outcomes.

## 6. Conclusions

The leadership framework of participative leadership has garnered extensive empirical focus underpinned by its generalizability and applicability to diverse organizational frameworks. The concept of participative leadership is underscored by the fundamental belief of leadership to be openly and genuinely expressive of the thought processes to ensure the prioritized accomplishment of each subordinate. Such an approach leads to work outcomes that are ethical and positive. This paper explored and evaluated participative leadership, focusing specifically on the institutional context trends evident within the organizations with a larger perspective on the broader organizational premises. The report offers initial insight into the positive intersection between employee loyalty and participative leadership that is reduced because of the impact of institutional context, thereby adversely affecting subordinates' performances.

In combination, the study infers that institutional context degrees depict variable operational functionalities of participative leaders. Thus, indicating that leaders and subordinates are distanced from each other because of the influence of institutional context. Furthermore, institutional context affects the evident behavior types of leadership, thereby manifesting challenges in aligning the decisions with the prioritizing needs of the subordinates. Therefore, the paper will serve as a significant scholarly reference to gain insight into the effects and impacts of institutional context on the association and interaction between participative leaders and their subordinates.

**Author Contributions:** Conceptualization, O.K. and H.E.; methodology, O.K. software, O.K. and H.E.; validation, O.K. and H.E; formal analysis, O.K. and H.E; investigation, O.K. and H.E.; resources, O.K. and H.E.; data curation, O.K. and H.E; writing—original draft preparation, O.K. and H.E.; writing—review and editing, O.K. and H.E.; visualization, O.K. and H.E.; supervision, O.K.; project administration, O.K. All authors have read and agreed to the published version of the manuscript.

**Funding:** This research received no external funding.

**Institutional Review Board Statement:** The study was conducted in accordance with the Declaration of Helsinki, and approved by the or Research Ethics Committee of University of Fort Hare's (protocol code MUR 39 1SZIN65).

**Informed Consent Statement:** Informed consent was obtained from all subjects involved in the study.

**Data Availability Statement:** Not applicable.

**Conflicts of Interest:** The authors declare no conflict of interest.

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
