# Peer review of "The Effect of Participative Leadership Style on Employees’ Performance: The Contingent Role of Institutional Theory"

_admsci, doi:10.3390/admsci12040195_

Round 1

Reviewer 1 Report

Dear Author(s),

After reading your manuscript, I must say that I find it very intriguing and well written. 
A few minor suggestions though are in order:

1. Please check the Journal Reference List and Citations Style Guide and apply the correct citation form for multiple authors accordingly.

2. Please clarify the instruments and the scales used.

3. Please show more clearly whether the Hypotheses are validated or not.

Looking forward to hearing from you!

Best regards!

Author Response

After reading your manuscript, I must say that I find it very intriguing and well written. 
A few minor suggestions though are in order:

  1. Please check the Journal Reference List and Citations Style Guide and apply the correct citation form for multiple authors accordingly.

Authors’ response:

Thank you very much for this great comment. We revised the citations to according to the guideline. Also, the journal team will assist and guide us during the production stage.

  1. Please clarify the instruments and the scales used.

Authors’ response:

This is now added to the methodology. Please see page 5 and 6 (The survey included the leading performance indicators (e.g., the speed of getting the task done, the customer satisfaction level, the number of functions, and complaints) ……….

  1. Please show more clearly whether the Hypotheses are validated or not.

Authors’ response:

This is now clarified. Please see page 7. For example, according to the findings, the relationship between participative leadership and employee loyalty was significantly ……….

Reviewer 2 Report

The paper is interesting and the topic can contribute to the theory and practice. 

More recent studies can be added to the paper to make it more critical. 

The discussion part must include some more discussions by comparing the current finding with the previous studies. 

Author Response

The paper is interesting and the topic can contribute to the theory and practice. 

Authors’ response:

Thank you very much for this comment. That is highly appreciated.

More recent studies can be added to the paper to make it more critical. 

Authors’ response:

Thanks for the great comment. More recent references were added.

For example,

Aksom, H., & Tymchenko, I. (2020). How institutional theories explain and fail to explain organizations—Journal of Organizational Change Management.

Budak, O., & Erdal, N. (2022). THE MEDIATING ROLE OF BURNOUT SYNDROME IN TOXIC LEADERSHIP AND JOB SATISFACTION IN ORGANIZATIONS. The South East European Journal of Economics and Business, 17(2), 1-17.

Chang, Y. Y., Chang, C. Y., Chen, Y. C. K., Seih, Y. T., & Chang, S. Y. (2021). Participative leadership and unit performance: evidence for intermediate linkages. Knowledge Management Research & Practice, 19(3), 355-369.

The discussion part must include some more discussions by comparing the current finding with the previous studies.

Authors’ response:

Thanks for the great comment. We improved the discussion part. We hope you find it satisfactory now, Please see page 9 and 10.

Reviewer 3 Report

Dear authors,

It was a pleasure reading your manuscript. I appreciate the effort invested into the research. Nonetheless, I have several questions and comments that should be considered before the paper is ready for publication.

1) Literature review - consideration of previous works:

    I very much miss the works of Meyer, Rowan, Di Maggio who are defining institutional theory.

More importantly is miss the critical discussion of institutional theory which has been discussed in various works. Eg "How institutional theories explain an fail to explain organisations" (Akson & Tymchenlo, 2020).

2) Sampling / Research design:

Your methodology misses to explain from which sectors the sample companies were from, how and why they were selected by the researcher(s). Reason for that is, that there are industry specifics for different industries, which cannot be generalised.

3) In your survey and further in your analysis you inquire the "overall performance" of employees. You lack to provide a definition for that and I assume that the understanding for each participant is different. When doing a survey with given terms, the boundaries of the terms have to be defined that all participants have the same understanding. Which were the key indicators for "overall performance"?

3) Your control variables are not very strong

4) Your research does not consider cultural differences that lead to a different leadership style as well. You only consider the institutional context. Your research does not identify cultural differences (eg. expat managers) or considers the impact of power distance on your research outcome.

Your research should at least highlight that the results were achieved in a very specific cultural environment and might not be able to be easily transferred.

With the comments above the current state of the manuscript reflects a research that missed to consider necessary factors which can lead to wrong results.

The manuscript lacks a statement about informed consent of the participants.

Author Response

Dear authors,

It was a pleasure reading your manuscript. I appreciate the effort invested into the research. Nonetheless, I have several questions and comments that should be considered before the paper is ready for publication.

1) Literature review - consideration of previous works:

    I very much miss the works of Meyer, Rowan, Di Maggio who are defining institutional theory.

More importantly is miss the critical discussion of institutional theory which has been discussed in various works. Eg "How institutional theories explain an fail to explain organisations" (Akson & Tymchenlo, 2020).

Authors’ response:

Thanks for the great comment. We improved the LR part and we added the two important studies you suggested. We hope you find it satisfactory now, Please see page 3.

2) Sampling / Research design:

Your methodology misses to explain from which sectors the sample companies were from, how and why they were selected by the researcher(s). Reason for that is, that there are industry specifics for different industries, which cannot be generalised.

Authors’ response:

Thanks for this valuable comment. We revised the methodology section accordingly. Please see page 5.

3) In your survey and further in your analysis you inquire the "overall performance" of employees. You lack to provide a definition for that and I assume that the understanding for each participant is different. When doing a survey with given terms, the boundaries of the terms have to be defined that all participants have the same understanding. Which were the key indicators for "overall performance"?

Authors’ response:

This is now added to the methodology. Please see page 5 (The survey included the overall performance indicators (e.g., the speed of getting the task done, the customer satisfaction level, the number of functions, and complaints) ……….

3) Your control variables are not very strong

Authors’ response:

We agree with you. We included the “Years of experience” to make it stronger.  

4) Your research does not consider cultural differences that lead to a different leadership style as well. You only consider the institutional context. Your research does not identify cultural differences (eg. expat managers) or considers the impact of power distance on your research outcome.

Your research should at least highlight that the results were achieved in a very specific cultural environment and might not be able to be easily transferred.

With the comments above the current state of the manuscript reflects a research that missed to consider necessary factors which can lead to wrong results.

Authors’ response:

That is really great comment. Due to the difficulty to add more about the cultural differences, we added this to the recommendations for future studies (see page 11). We highlighted that our results might be applied to specific cultural environment and might not be able to be easily transferred (see page 10 - In addition, the results were achieved in a particular …).

The manuscript lacks a statement about informed consent of the participants.

Authors’ response:

Thanks for highlighting this issue. We added this statement to the methodology. Please see page 5 - An electronic medium was used for data collection to share the questionnaires during work hours, and informed consent was included in the questionnaire).

Round 2

Reviewer 3 Report

Dear author(s),

Thank you very much for the revised version of your manuscript and the improvements you implemented. The current version is in a form that is ready for puplication.